# A Veritable Zoology of Successive Phase Transitions in the Asymmetric *q*-Voter Model on Multiplex Networks

**DOI:** 10.3390/e22091018

**Published:** 2020-09-11

**Authors:** Anna Chmiel, Julian Sienkiewicz, Agata Fronczak, Piotr Fronczak

**Affiliations:** Faculty of Physics, Warsaw University of Technology, Koszykowa 75, PL-00-662 Warsaw, Poland; julian.sienkiewicz@pw.edu.pl (J.S.); agata.fronczak@pw.edu.pl (A.F.); piotr.fronczak@pw.edu.pl (P.F.)

**Keywords:** multiplex networks, voter model, opinion dynamics, successive phase transition, hybrid phase transition

## Abstract

We analyze a nonlinear *q*-voter model with stochastic noise, interpreted in the social context as independence, on a duplex network. The size of the lobby *q* (i.e., the pressure group) is a crucial parameter that changes the behavior of the system. The *q*-voter model has been applied on multiplex networks, and it has been shown that the character of the phase transition depends on the number of levels in the multiplex network as well as on the value of *q*. The primary aim of this study is to examine phase transition character in the case when on each level of the network the lobby size is different, resulting in two parameters $q_{1}$ and $q_{2}$. In a system of a duplex clique (i.e., two fully overlapped complete graphs) we find evidence of successive phase transitions when a continuous phase transition is followed by a discontinuous one or two consecutive discontinuous phase transitions appear, depending on the parameter. When analyzing this system, we even encounter mixed-order (or hybrid) phase transition. The observation of successive phase transitions is a new quantity in binary state opinion formation models and we show that our analytical considerations are fully supported by Monte-Carlo simulations.

## 1. Introduction

It is pretty obvious that modeling opinion dynamics [1,2,3,4,5,6] is a tricky task that can be seen as maneuvering between two distinct extremes. On one side, there are classical binary state models [7] that are often subject to exact analytical treatment, although their assumptions and formulation can be seen as oversimplified, especially from the social science point of view. On the other hand, it is also possible to move towards a very general approach of agent-based models [8,9,10,11], equipping individuals with large vectors of attributes, applying detailed, very complex rules and, last but not least, taking into account multidimensional structure of interactions.

As underlined before [12], the so-called *q*-voter model with independence, understood as stochastic noise (later on in this paper referred as simply *q*-voter model for brevity), is of particular interest in the group of binary opinion models. The idea that *q*-lobby (a group of *q* nodes chosen form of all the neighbors of an agent) acting on an individual needs to be unanimous in order to change agent’s opinion has solid grounds in social sciences. As a seminal example one can refer to Asch experiments [13] that raise the importance of the agreement among a group that is bound to convince a participant to change his/her opinion. The size of the group (lobby) and the homogeneity of the state is also particularly essential as it was shown in Stanley Milgram’s studies [14]. In this way, despite being relatively simple, *q*-voter model can be said to have realistic assumptions. From the macroscopic perspective the *q*-voter model shows how, owing to the introduction of the noise to the system, society is going to a consensus (i.e., one opinion dominates) which is an example of a spontaneous symmetry breaking.

However, *q*-voter model with independence manifests also another property that is claimed to be very often observed in social systems [15,16]: a hysteresis, i.e., the dependence of the current state of the system on previous ones. This phenomenon is directly connected to an especially interesting physical concept of a discontinuous phase transition. We need to stress that although a phase transition between the ordered and disordered state is typically observed in the vast majority of binary-state models such as Sznajd model [2], voter model [3], and threshold model [6], it is usually a continuous phase transition. The fact that *q*-voter model with independence, in spite of its relative simplicity, displays the discontinuous phase transition driven by a stochastic noise makes it a very attractive playground. When examined on the topology of a complete graph (i.e., a clique) which is subject to mean-field approach, the transition changes its character from continuous to discontinuous for q≥6 [17]. It is also possible to examine the *q*-voter model with independence for more realistic network topologies, such as Erdős–Rényi random graphs, Barabasi–Albert evolving networks, or Watts–Strogatz model using pair approximation [18], however in the limiting case, these solutions coincide with the complete graph one.

The above-mentioned graph structures can be criticized as being oriented on just a single aspect of one’s relationship. In reality, an individual is subject to constant pressure from different groups that create separate networks—such a situation is especially evident in the case of different social on-line media. Exchanging comments with a friend, let us say, on Twitter, does not necessarily mean that the same person is among one’s contacts in Facebook. In the last decade, multiplex networks [19] were proposed as an elegant tool to model such aspects of activity and, without any doubt, they have become one of the most active areas of recent network research mainly due to the fact that many real-world systems possess layers in a natural way [20]. A lot of attention has also been devoted to the analysis of various dynamics on multiplex networks, including diffusion processes [21,22], epidemic spreading [23], majority-vote [24], and voter dynamics [25,26]. The *q*-voter model examined on a duplex (i.e., consisting of two levels) networks as well as on an arbitrary number *L* of layers [12] brings interesting results: the value of *q* for which the transition changes its character from continuous to discontinuous moves from q≥6 observed for a monoplex (i.e., L=1) to q≥5 for a duplex. In case of L≥3, this critical value becomes constant and equal to 4—so far there is no straightforward explanation of these phenomena. Lately, also the pair approximation technique has been used to examine different topologies [27].

The major drawback of such a setting of the *q*-voter model on multiplex networks that we want to tackle in this paper lays in its symmetry, i.e., the lobby acting on an individual on each level has the same size *q*. Such an assumption does not seem to be justified as it is rather clear that we pay more attention to some groups while almost neglecting others—if a person is less devoted to on-line groups than to real-life friends, even a smaller set chosen from the latter will affect him/her stronger than larger group recruiting from the first ones. To overcome these issues we introduce in this study an asymmetric *q*-voter model with independence on a duplex clique where lobby sizes on different levels are described by parameters $q_{1}$ and $q_{2}$. This simple step brings unexpected and interesting results from the statistical physics point of view: for certain values of $q_{1}$ and $q_{2}$, we observe so-called successive phase transitions [28]. In fact, depending on the actual value of $q_{1}$ and $q_{2}$, the model gives two consecutive discontinuous phase transitions or a continuous phase transition following a discontinuous one.

The rest of the paper is organized in the following way: in Section 2, we first introduce in detail the *q*-voter model on a single-layer network, i.e., a monoplex, giving rationale for the rate equations used. In particular we pay attention to the complete graph case but also describe briefly the results obtained recently for more sophisticated topologies. Then, we move to *q*-voter model on a duplex clique, underlining the way dynamical rules take into account the existence of more than one layer. The section ends with the description of the asymmetric *q*-voter model with independence. Section 3 comes back to the symmetric model, presenting a different way to describe the phase diagram than it has been done in [12] and this approach is then used to display the results of the asymmetric model in the remaining part of the section. The outcomes from the model and their consequences are discussed in Section 4, that also summarizes the paper.

## 2. q-Voter Model on Single- and Multilayer Networks

### 2.1. The q-Voter Model on a Single Monoplex Clique

Let us briefly describe the *q*-voter model with independence on a monoplex complete graph [17]. In such a setting, we consider a set of *N* individuals, which are represented by binary variables $S_{i}=\pm 1$ (spins ’up’ or ’down’). At each elementary time step $\Delta_{t}$ we randomly choose an *i*-th node (i.e., a voter) and its so-called *q*-lobby, which is a randomly picked group of *q* individuals. Only the self-consistent *q*-lobby can act on the voter. With probability 1−p, the *q*-lobby (provided it is homogeneous) exerts influence on the state of the voter, which means that the voter flips its state to the state of the *q*-lobby. On the other hand, with probability *p*, the voter behaves independently—with equal probabilities it changes its state to the opposite direction $S_{i}\left(t+\Delta_{t}\right)=-S_{i}\left(t\right)$ or keeps its original state, i.e., $S_{i}\left(t+\Delta_{t}\right)=S_{i}\left(t\right)$. In a single time step $\Delta_{t}=\frac{1}{N}$, there are three scenarios possible—the number of up-spins $N_{↑}\left(t\right)$ will either increase by 1, decrease by 1, or remain constant. As a consequence, the concentration $c\left(t\right)=\frac{N_{↑}\left(t\right)}{N}$ increases or decreases by $\frac{1}{N}$ or remains constant according to the formulas
(1)$$\begin{array}{ccc} \gamma^{+}\left(c\right) & = & \mathrm{Pr}\left\{c\left(t+\Delta_{t}\right)=c\left(t\right)+\frac{1}{N}\right\},\\ \gamma^{-}\left(c\right) & = & \mathrm{Pr}\left\{c\left(t+\Delta_{t}\right)=c\left(t\right)-\frac{1}{N}\right\},\\ \gamma^{0}\left(c\right) & = & \mathrm{Pr}\left\{c\left(t+\Delta_{t}\right)=c\left(t\right)\right\}=1-\gamma^{+}\left(c\right)-\gamma^{-}\left(c\right) \end{array}$$
that describe the probabilities of the change of concentration. The time evolution of the average concentration is then given by the following rate equation:(2)$$c\left(t+\Delta_{t}\right)=c\left(t\right)+\frac{1}{N}\left[\gamma^{+}\left(c\right)-\gamma^{-}\left(c\right)\right].$$
Let us underline here that, when we deal with large systems N≫1, in particular when N→∞ the time interval $\Delta_{t}$ goes to zero, giving in result
(3)$$\frac{\partial c}{\partial t}=\gamma^{+}\left(c\right)-\gamma^{-}\left(c\right),$$
where $\gamma^{+}\left(c\right)$, $\gamma^{-}\left(c\right)$ are probabilities that a single voter changes its state, respectively, from −1 to 1 or from 1 to −1 and can be written as
(4)$$\begin{array}{c} \gamma^{+}\left(c\right)=\left(1-p\right)\left(1-c\right)c^{q}+\frac{p(1-c)}{2},\\ \gamma^{-}\left(c\right)=\left(1-p\right)c{\left(1-c\right)}^{q}+\frac{pc}{2}. \end{array}$$
The first component in both equations is related to the conformity behavior of the voter and the second one appears as a result of the independent behavior. The voter flips its state due to conformity rule only when the *q*-lobby is homogeneous, i.e., all the chosen neighbors have the same state. The chance for such a situation to occur is proportional to the level of concentration *c*. Let us consider in detail $\gamma^{+}\left(c\right)$ here: in the first component, 1−p is the chance of applying the conformity rule and 1−c is the chance to randomly pick up a voter with state −1 while $c^{q}$ gives the probability to find a *q*-lobby consisting of voters with state 1. In the second component, *p* gives the probability that the voter behaves independently, 1−c is the chance to randomly pick up a voter with state −1 and 1/2 is the chance that a voter would flip to the opposite state 1.

The macroscopic behavior of such a system can be described by its magnetization:(5)$$m\left(t\right)=\frac{1}{N}\sum_{i=1}^{N}S_{i}\left(t\right)=\frac{N_{↑}\left(t\right)-N_{↓}\left(t\right)}{N}.$$
with $N_{↓}\left(t\right)$ being the number of down-spins at time *t* and $N=N_{↑}\left(t\right)+N_{↓}\left(t\right)$ for any *t*. On the other hand, magnetization m(t) is directly related to the concentration c(t) by
(6)m(t)=2c(t)−1
The probabilities $\gamma^{+}\left(c\right)$ and $\gamma^{-}\left(c\right)$ can be easily rewritten in the magnetization-dependent forms $\gamma^{+}\left(m\right)$ and $\gamma^{-}\left(m\right)$:(7)$$\begin{array}{c} \gamma^{+}\left(m\right)=\left(1-p\right)\frac{1-m}{2}{\left(\frac{1+m}{2}\right)}^{q}+\frac{p}{4}\left(1-m\right),\\ \gamma^{-}\left(m\right)=\left(1-p\right)\frac{1+m}{2}{\left(\frac{1-m}{2}\right)}^{q}+\frac{p}{4}\left(1+m\right). \end{array}$$

In order to find the stationary state, the following requirement for the effective force has to be fulfilled:(8)$$F\left(m\right)=\gamma^{+}\left(m\right)-\gamma^{-}\left(m\right)=0.$$
However, the above quantity does not allow to judge upon the stability of the solutions. To acquire such information, one needs to integrate the effective force F(m) obtaining the effective potential
(9)V(m)=−∫F(m)dm.
Similar to the Landau theory, global minimum of the effective potential V(m) gives the stable stationary solution, local minima are metastable solutions and the maximum of V(m) is related to unstable solutions.

It has been shown [17] that the system, described by the *q*-voter model with independence, undergoes the phase transition at $p=p_{c}\left(q\right)$. For $p<p_{c}$, the majority coexists with the minority opinion (ordered state) and for $p>p_{c}$ there is a status-quo (disordered state). Interestingly, for q≤5 the phase transition is continuous, whereas for q>5 it becomes discontinuous.

### 2.2. The q-Voter Model on a Monoplex Network

The *q*-voter model with a stochastic noise arising from independence was precisely investigated on a set of complex networks in a recent work [18]. Owing to the application of the so-called pair approximation method, it was possible to find a comprehensive mathematical description of the model behavior on several complex structures including Erdős–Rényi random graphs, Barabási–Albert evolving networks, and the Watts–Strogatz model, as well as on a random regular graphs. Analytical solutions presented in [18] are in a very good agreement with Monte Carlo simulation, especially for networks with small clustering coefficient and for large average degree 〈k〉 values. The character of the phase transition changes from continuous to discontinuous when *q* becomes larger than 5, which is the same as in complete graph case [17]. This observation means that the structure of networks has the influence only on the critical value of the noise parameter $p_{c}$ but not on the character of the phase transition.

### 2.3. The Symmetric q-Voter Model on a Duplex Clique

Let us now introduce the definition of a duplex clique, which is a particular case of a multiplex [19]. A duplex clique is a network that consists of two distinct levels (layers), each of which is represented by a complete graph (i.e., a clique) of size *N*. Levels represent two different communities (e.g., Facebook and a school class), but are composed of exactly the same people—each node possesses a counterpart node in the second level. Such an assumption reflects the fact that we consider fully overlapping levels, it is an idealistic scenario. We also assume that each node possesses the same state on each level, which means that the society consists of non-hypocritical individuals only. Due to this feature, we can simplify our analysis by considering concentration c(t) only on one level. However, we need to stress that the changes of the state of the node occur under the influence of both levels.

In this paper, we analyze an asymmetric *q*-voter model which is an extension of the original symmetric case on a duplex network defined previously in [12]. Among the three presented methods that transfer the model from a monoplex to multiplex network [12], the rule called LOCAL&AND seems to be most promising from the point of statistical physics and also produce qualitatively new behavior with respect to monoplex structure. We briefly describe the LOCAL&AND rule on the duplex clique. The independence in this approach is LOCAL, i.e., the dynamics runs separately on each level. A voter is independent on the first level with probability *p* and with probability 1−p behaves as a conformist—it is under the influence of the *q*-lobby on this level. The same situation is on the second level, where, regardless of the first level we choose if the voter behaves independently or conforms to the *q*-lobby on the second level. Finally, we change the state of the voter only when both separated dynamics give in result the same state which is an equivalent of an AND logical rule. Exact formulas for probabilities $\gamma^{+}\left(c\right)$ and $\gamma^{-}\left(c\right)$ in the case of LOCAL&AND on the duplex clique read
(10)$$\begin{array}{cc} \gamma^{+}\left(c\right)= & {\left(1-p\right)}^{2}\left(1-c\right)c^{2q}+p\left(1-p\right)\left(1-c\right)c^{q}+\frac{p^{2}\left(1-c\right)}{4},\\ \gamma^{-}\left(c\right)= & {\left(1-p\right)}^{2}c{\left(1-c\right)}^{2q}+p\left(1-p\right)c{\left(1-c\right)}^{q}+\frac{p^{2}c}{4}. \end{array}$$
There are three factors appearing in the above equations: the first is related to the situation when on both levels the voter behaves as a conformist, the second one is a mixed factor (the voter is a conformist on one of the levels and on the second it acts independently), and finally the last is a result of two independent behaviors of the voter. Just like in the previously described analysis for the monoplex network, we define the effective force and the effective potential dependent on the magnetization. Numerical analysis of effective force (Equation 8) and the effective potential (Equation 9) allows us to create a detailed phase diagram of the examined system.

It was shown [12] that qualitative changes in the phase transitions can be observed for the LOCAL&AND rule—for the duplex clique, the phase transition becomes discontinuous for q=5, whereas for a monoplex such behavior is observed for q≥6.

### 2.4. Asymmetric q-Voter Model on Duplex Networks

In the asymmetric *q*-voter model, the size of the *q* lobby can be different for each level: we introduce parameter $q_{1}$ reflecting the size of the lobby on the first level and $q_{2}$ on the second one. Exact formulas for probabilities $\gamma^{+}\left(c\right)$ and $\gamma^{-}\left(c\right)$ in the case of LOCAL&AND on the duplex clique can be written as follows: (11)$$\begin{array}{cc} \gamma^{+}\left(c\right)= & \left(1-c\right)\left[{\left(1-p\right)}^{2}c^{q_{1}+q_{2}}+\frac{p}{2}\left(1-p\right)c^{q_{1}}+\frac{p}{2}\left(1-p\right)c^{q_{2}}+\frac{p^{2}}{4}\right],\\ \gamma^{-}\left(c\right)= & c\left[{\left(1-p\right)}^{2}{\left(1-c\right)}^{q_{1}+q_{2}}+\frac{p}{2}\left(1-p\right){\left(1-c\right)}^{q_{1}}+\frac{p}{2}\left(1-p\right){\left(1-c\right)}^{q_{2}}+\frac{p^{2}}{4}\right]. \end{array}$$

## 3. Results

### 3.1. Symmetric q-Voter Model on Duplex Networks

In this section, we revisit the symmetric *q*-voter model on duplex networks, analyzing it in a slightly different way than originally presented in [12]. To find a stationary solution we numerically solve Equation (Equation 8) and analyze the phase diagram and stability using the effective potential (Equation 9), distinguishing between stable and metastable solutions. The approach introduced here will be our tool to study the asymmetric *q*-voter model in the next section.

In the diagram shown in Figure 1, the solid line indicates phase transitions which are continuous for q<Q (marked in blue) and discontinuous for q>Q (marked in red). The continuous transition changes its character to discontinuous for q=Q≃4.5. The dashed lines represent spinodals that accompany discontinuous phase transitions.

The phase diagram area is divided into two parts: gray (which corresponds to the ordered phase of the system, |m|>0) and white (in which the system is disordered, |m|=0). The (right) hatched area between spinodals is called the coexistence region. When the state parameters (i.e., *q*—the clique size and *p*—the level of independence) belong to this area, the system can be observed in two stationary states, one of which is stable and the other is metastable. The stable state corresponds to the global minimum of the potential V(m) (see Figure 2), while the metastable state corresponds to its local minimum. In the hatched white area (between the discontinuous transition line and the upper spinodal), the disordered phase is stable. In the hatched gray area (between the transition line and the lower spinodal), the stable phase is the ordered one and the disordered phase is metastable.

In Figure 2, for two values of *q* it is shown how magnetization of the system changes as the parameter *p* increases. Red bold solid lines represent stable states of the system while thin red lines stand for metastable states. Blue dashed lines indicate unstable solutions of Equation (Equation 8). To the right of this figure, there are also auxiliary charts showing how the potential of the system V(m) given by Equation (Equation 9) looks like for selected values of *p*. It is easy to see that stable solutions (bold red lines) always correspond to the global minima of V(m), metastable solutions (thin red lines) are visible as its local minima, and, finally, unstable states (blue dotted lines) coincide with the maxima of V(m).

It is worth to stress that although one usually considers *q*-lobby size as an integer value, Equations (Equation 8) and (Equation 9) can be also solved for a non-integer value of *q*. Similarly it is possible to obtain non-integer values of *q* in numerical simulations by assigning probability distribution of *q* as it was done in [29].

Finally, let us also note that, in general, a typical way (Landau approach) to examine the stability of the solutions is to approximate V(m) with a suitable polynomial (usually of order 4 or 6) and obtain results for critical points. However, it has been shown that even for relatively simple systems [30] this analysis might not bring the expected outcomes. Moreover, due to high complexity of the problem (high values of $q_{1}$ and $q_{2}$) one would need to use high orders of polynomials, making it hard to evaluate in an analytical way. Instead, as mentioned before, we numerically examine V(m) to find the character of the solutions of F(m)=0.

### 3.2. Asymmetric q-Voter Model on Duplex Networks

In the case of the asymmetric *q*-voter model on the duplex clique, we follow a similar approach solving equation for effective force F(m)=0 and analyzing the behavior of the effective potential to find the character of the observed phase transitions. The important feature of the asymmetric model is the fact that we can study different lobby sizes on each of the level, which brings the model closer to real-world situations. However, both levels are identical and indistinguishable, i.e., we obtain the same results when the values of $q_{1}$ and $q_{2}$ are swapped. This symmetry is clearly visible in Figure 3 where we show the character of the phase transition of the system for small values of $q_{1}$ and $q_{2}$. If on both levels q≤4, only continuous phase transition takes place, which is in agreement with the intuition from the previous analysis of the symmetric *q*-voter model. However, if $q_{1}\leq 4$ and $q_{2}\geq 5$, the phase transition changes its character. On the other hand, when on one of the levels the *q*-lobby is equal to 5 we find a much more complex picture where first a discontinuous then continuous and finally once again discontinuous phase transition is observed with increasing $q_{2}$ (see also the inset in Figure 4); for $q_{1}=6$ only discontinuous phase transition is present in the examined region (see Figure 5). Let us underline, however, it is for larger values of $q_{2}$ that one can observe a very interesting set of phenomena occurring in the model. In the following sections we shall focus on two specific values of $q_{1}$ ($q_{1}=5$ and $q_{1}=6$) and present a detailed study of the phase diagram in such cases, noticing the presence of successive phase transitions.

#### 3.2.1. The $q_{1}=5$ Case

In the phase diagram shown in Figure 4, there are three main areas which are separated by solid lines (red and blue) and marked with different colors: dark gray, light gray, and white. The hatched and dotted areas between spinodals indicate various coexistence regions. The homogeneous white area represents disordered (with |m|=0) states of the system. Gray areas (dark and light) stand for the ordered phases: $\left|m_{1}\right|>\left|m_{2}\right|>0$, respectively. Blue solid lines are continuous (second order) phase transitions lines. In the system studied, such continuous transitions can be observed for two different ranges of the $q_{2}$ parameter. In particular, when $q_{2}\in \left(A,B\right)$ the transition occurs in a way that is similar as in the symmetric system for q<Q (cf. Figure 1). For $q_{2}>C$, a succession of phase transitions can be observed when the parameter *p* increases. The first order transition (red line) between two ordered phases, $\left|m_{1}\right|\to \left|m_{2}\right|$, is followed by the second order transition (blue line), $\left|m_{2}\right|\to 0$. For $q_{2}\in \left(C,D\right)$, the continuous transition occurs in the region of coexistence that accompanies the discontinuous transition $\left|m_{1}\right|\to \left|m_{2}\right|$, in which the more ordered between two phases with non-zero magnetization is metastable. Finally, for $q_{2}>D$, the continuous transition occurs in a similar way as for $q_{2}\in \left(A,B\right)$. The transition point $q_{2}=C≃25$ at which the line of continuous transition intersects the discontinuous line is of particular interest. At this point, the disordered phase |m|=0 changes its character from metastable to stable and the mixed-order (or hybrid) transition takes place. This interesting phase transition [31,32] consists of a step change in magnetization which occurs simultaneously with diverging fluctuations, when the transition point is approached from higher values of *p*. Finally, let us emphasize that the hybrid transition point $q_{2}=C$ divides the discontinuous transition line into two parts. For $q_{2}\in \left(B,C\right)$, when the parameter *p* increases, the transition occurs between the ordered and disordered phases, $\left|m_{1}\right|\to 0$, while for $q_{2}>C$ it is between two ordered phases, $\left|m_{1}\right|\to \left|m_{2}\right|$.

In Figure 5, for selected values of $q_{2}$, it is shown how magnetization of the system changes as the parameter *p* increases. We use the same way of marking the type of solution as in case of Figure 2: red bold solid lines represent stable states, thin red lines stand for metastable states, and blue dashed lines indicate unstable solutions of Equation (Equation 8). To the right of each panel in this figure, there are also additional charts showing what the potential of the system V(m) given by Equation (Equation 9) looks like for specific values of *p*. In Figure 5c, we observe a mixed (hybrid) phase transition indicated by a flat region in the potential V(m) (marked by number “2” in a circle). Our analysis is in agreement with Monte Carlo simulations, see Appendix A for details.

#### 3.2.2. The $q_{1}=6$ Case

In the same manner as in the case of $q_{1}=5$ in Figure 6, we show the phase diagram for $q_{1}=6$. Contrary to the $q_{1}=5$ example, here only the first order transition (red line) between two ordered phases is observed. For $q_{2}\in \left(1,E\right)$ single discontinuous phase transition between an ordered ($\left|m_{1}\right|>0$) and disordered state (|m|=0) appears—the inset panel on the left-hand side in Figure 6 shows details for small values of $q_{2}$, where in the case of $q_{1}=5$ a continuous phase transition is visible. The right-hatched areas between spinodals indicate coexistence regions. The homogeneous white area represents disordered (with |m|=0) states of the system. The gray areas (dark and light) stand for the ordered phases: $\left|m_{1}\right|>\left|m_{2}\right|>0$, respectively. For $q_{2}=E$, a metastable solution $\left|m_{2}\right|>0$ appears and one discontinuous phase transition $\left|m_{1}\right|\to \left|m_{2}\right|$ is followed by another discontinuous $\left|m_{2}\right|\to 0$. Thus, similar as in the $q_{1}=5$ case, we have successive phase transitions, however this time both are first-order type, in effect. From $q_{2}=F$ the state $\left|m_{2}\right|>0$ becomes stable, in result, there are as many as three areas of coexistence regions: (i) right-hatched area between $\left|m_{1}\right|>0$ and |m|=0, (ii) left-hatched area between $\left|m_{2}\right|>0$ and |m|=0, and (iii) dotted area between $\left|m_{1}\right|>0$ and $\left|m_{2}\right|>0$. Inset panel on the right-hand side of Figure 6 magnifies the region of phase coexistence and allows closer inspection of this exotic behavior: in particular, for certain values of $q_{2}$ and *p* one observes that three phases coexist. This phenomenon can also be studied using Figure 7, where four specific cases of $q_{2}$ have been selected to show the behavior of the average magnetization on noise parameter *p*. Contrary to the $q_{1}=5$ case, there is no evidence of a hybrid phase transition for any values of $q_{2}$ and *p*. Similar to $q_{1}=5$, also in this case our analysis is in agreement with Monte Carlo simulations (see Appendix A).

### 3.3. Limiting Behavior

As mentioned before, the overall form of Equation (Equation 8) is rather complex and closed-form solutions are possible only for small values of $q_{1}$ and $q_{2}$. In other cases we need to use semi-analytical or numerical methods. It is, however, fairy simple to obtain an analytical formula for the point $p=p^{*}$
where the disordered solution |m|=0 changes its character from unstable to stable (or metastable). It follows that $p^{*}$ can be obtained from the condition
(12)$${\left|\frac{\partial F(c)}{\partial c}\right|}_{c=\frac{1}{2}}=0,$$
which gives
(13)$$p^{*}=\frac{2^{q_{2}}+\left(2^{q_{2}}-4\right)q_{1}+\left(2^{q_{1}}-4\right)\left(q_{2}-1\right)+\sqrt{4^{q_{2}}{\left(q_{1}-1\right)}^{2}+4^{q_{1}}{\left(q_{2}-1\right)}^{2}+2^{q_{1}+q_{2}+1}\left(q_{1}q_{2}+q_{1}+q_{2}-1\right)}}{2\left(2^{q_{2}}-2\right)q_{1}+\left(2^{q_{1}}-2\right)\left(2^{q_{2}}+2q_{2}-2\right)}$$
It is worth to mention here, that in the case where we assume a symmetric model (i.e., $q_{1}=q_{2}=q$) we arrive at
(14)$$p_{q_{1}=q_{2}}^{*}=\frac{2(2q-1)}{2\left(2q-1\right)+2^{q}}$$
which exponentially drops to 0 with increasing *q*.

Let us now check the value of $p^{*}$ for the asymmetric case, assuming that we keep $q_{1}$ constant and $q_{2}\to \infty$. Interestingly, in this limiting case we obtain that
(15)$$p_{q_{2}\to \infty }^{*}=\frac{q_{1}-1}{q_{1}-1+2^{q_{1}-1}}$$
which coincides with the value obtained by Nyczka et al. for the *q*-voter model on a monoplex network [17]. Although formally Equation (Equation 15) describes $q_{2}\to \infty$, it is easy to check that we arrive at $p_{q_{2}\to \infty }^{*}$ even for relatively small values of $q_{2}$. Inspecting the lower spinodal of $\left|m_{1}\right|\to 0$ and then the stable solution of $\left|m_{2}\right|>0$ in Figure 4, it is obvious that it stabilizes on the value of $p^{*}=1/5$ for $q_{2}\approx 15$. A similar situation can be spotted in the case of $q_{1}=6$ (cf. Figure 5), where the limiting value is $p^{*}=5/37$. The conclusion from these considerations is the following: if there is a significant difference between $q_{2}$ and $q_{1}$ ($q_{2}\gg q_{1}$), the system starts to behave as if it were a monoplex network described by parameter $q_{1}$. Of course, we can still observe the behavior characteristic for the asymmetric model, i.e., the succession of phase transitions, however the size of the first phase transition decreases with growing $q_{2}$, to disappear when $q_{2}\to \infty$. Figure 8 illustrates this behavior for $q_{1}=5$ and $q_{1}=6$.

## 4. Conclusions

Let us start these conclusions by addressing a maybe provocative title of this work, in particular the word “zoology” that might bring pejorative connotations. At the beginning of the paper we tried to draw a suggestive picture of two possible roads to modeling of opinion dynamics, distinguishing between binary state models and an agent-based approach. The crucial advantage of binary models was connected to their simplicity, which in turn can be seen as an important factor when it comes to describing specific social phenomena. The other advantage of a binary opinion model, such as the *q*-voter model with independence is that, at least in theory, it could be treated as a generic structure, i.e., one does not expect drastic changes in the observable when certain changes are introduced to the system.

This idea can be easily illustrated by comparing base *q*-voter model with independence [17] and the same model on a duplex clique with LOCAL&AND dynamics [12]: the major difference is a shift of the value of *q* for which the continuous phase transition becomes a discontinuous one (q=6 for monoplex and q=5 for duplex). One could naively expect the same situation while examining an asymmetric *q*-voter model, which could be treated as a generalization of the symmetric version. However, the analysis shown in this study, backed with Monte-Carlo simulations presents a different scenario: introduction of different lobby sizes on each level of the duplex clique dramatically change the description from the statistical physics point of view (see Table 1 for a comparison). Instead of a single first- or second-order phase transition, we observe now the phenomenon of successive phase transitions. Our detailed analysis shows that the most diverse behavior appears if $q_{1}=5$: (i) for $q_{2}=1$ a discontinuous phase transition, (ii) for $q_{2}\in \langle 2,3\rangle$ a continuous phase transition, (iii) for $q_{2}\in \langle 4,24\rangle$ again a discontinuous phase transition, (iv) for $q_{2}=25$ a mixed order phase transition, and (v) for $q_{2}>25$ successive phase transitions—a continuous phase transition following a discontinuous one. For $q_{1}=6$ (and larger values), we observe two different behaviors: (i) for $q_{2}<35$ a discontinuous phase transition and (ii) for $q_{2}>35$ successive phase transitions when two discontinuous transitions follow each other. Results also suggest that for very large differences between $q_{1}$ and $q_{2}$ the system can be described in the following way: the “main” phase transition is identical as in the case of the monoplex while the second transition is imposed on the first one and vanishes for $q_{2}\to \infty$. In other words, for sufficiently large values of $q_{2}$, the first level does not “feel” the second and behaves strictly as a monoplex clique.

Various extensions of the *q*-voter model were recently studied on monoplex networks like the threshold *q*-voter model [33] or noisy threshold *q*-voter model [34], where a minimal agreement of at least $q_{0}\leq q$ opposite opinions is sufficient to flip the central agent’s opinion. Although adding the $q_{0}$ parameter influences the character of the phase transition and enriches the phase diagram, the successive phase transitions are not observed. The reasons for the appearance of successive phase transitions in this class of models is a puzzle since it is obvious that just the multiplex structure cannot be responsible for this exotic behavior [12,27].

This is the zoology of phase transitions mentioned in the title—one needs to underline that in some sense it brings to the front the problem of social reliability of such models. Although the introduced change seems to be small, it is far from obvious how the observed phenomenon can be interpreted from the social sciences perspective.

So far, the phenomenon of successive phase transitions has been observed only in selected physical systems [28,35,36]. The behavior manifesting in the fact that for a selected set of parameters the system can be in one of the three phases is, at least according to our knowledge, a new quantity in the binary state opinion formation models. In a three-state opinion model [37], non-monotonic changes of global ordering (qualitatively similar to the phenomena of successive phase transitions) have recently been observed in modular network.

It is also not clear if extending the model into a higher number of layers brings additional exotic behavior, such as a cascade of phase transitions. A similar situation has been observed when the so-called *q*-Ising model was examined for monoplex [38] and partially duplex cliques [39]—also in this case, simply introducing overlapping cliques leads to a surprising result.

Finally, let us also underline that among some further plans it should be of use to consider network topology (e.g., random graphs or scale-free networks) instead of this very simple complete graph approach in a similar way as it has been done by pair approximation for monoplex [18] and symmetric duplex [27] graphs. Moreover, one needs to stress that the *q*-voter model with independence is just one specific extension of the basic *q*-voter model—it would be interesting to analyze also the other versions in search for the evidence of successive phase transitions. In this way it might be possible to uncover the mechanism of phenomenon in question in this class of models.

## Figures and Tables

**Figure 1 entropy-22-01018-f001:**
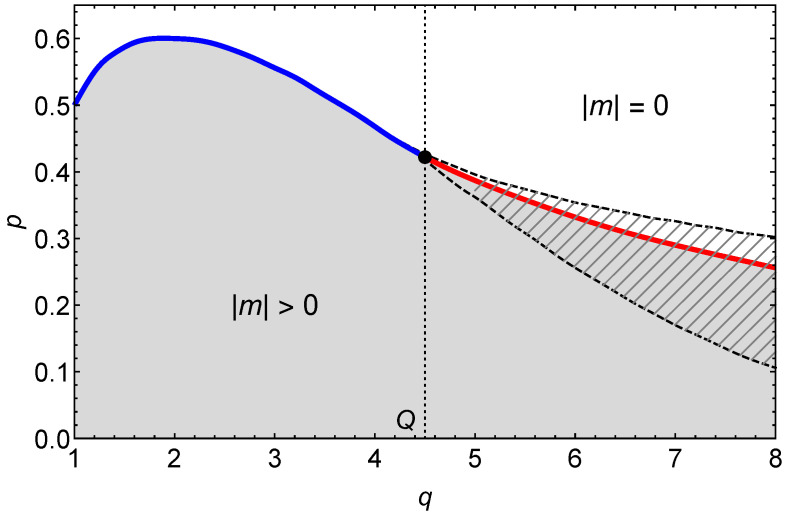
Phase diagram for the symmetric *q*-voter model on duplex clique with the LOCAL&AND rule.

**Figure 2 entropy-22-01018-f002:**
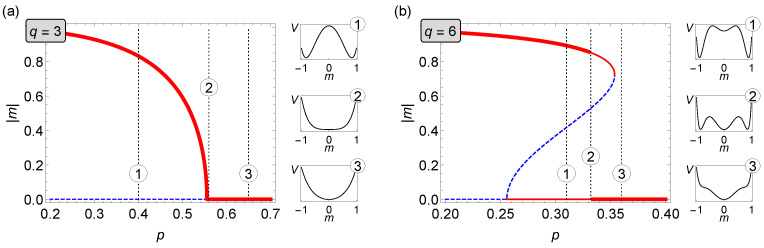
Average magnetization against the parameter *p* for two different lobby sizes in the symmetric voter model on duplex clique with the LOCAL&AND rule. (**a**) q=3 and (**b**) q=6 (cf. Figure 1).

**Figure 3 entropy-22-01018-f003:**
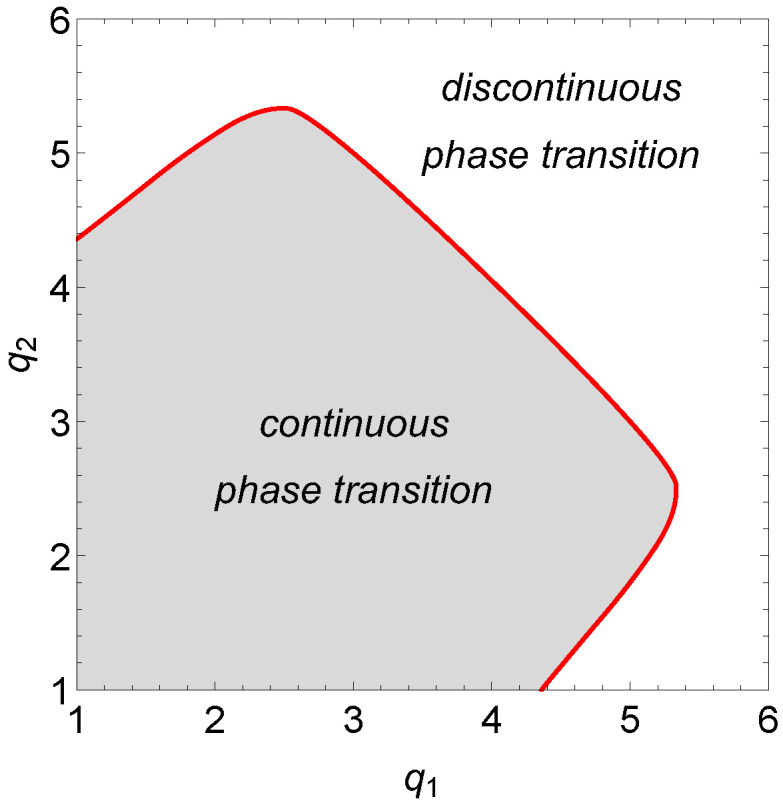
Visualization of the character of the phase transition for the asymmetric $\left(q_{1},q_{2}\right)$-voter model on duplex cliques with the LOCAL&AND rule for small values of *q*.

**Figure 4 entropy-22-01018-f004:**
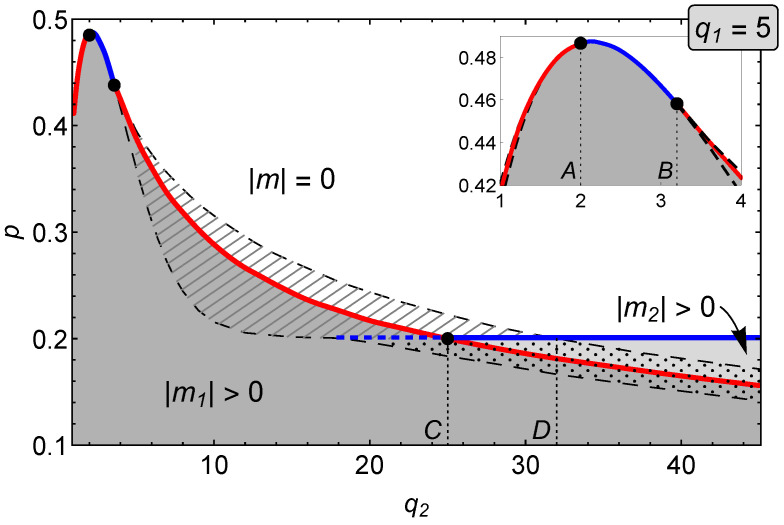
Phase diagram for the asymmetric $\left(q_{1},q_{2}\right)$-voter model on duplex cliques with the LOCAL&AND rule and $q_{1}=5$.

**Figure 5 entropy-22-01018-f005:**
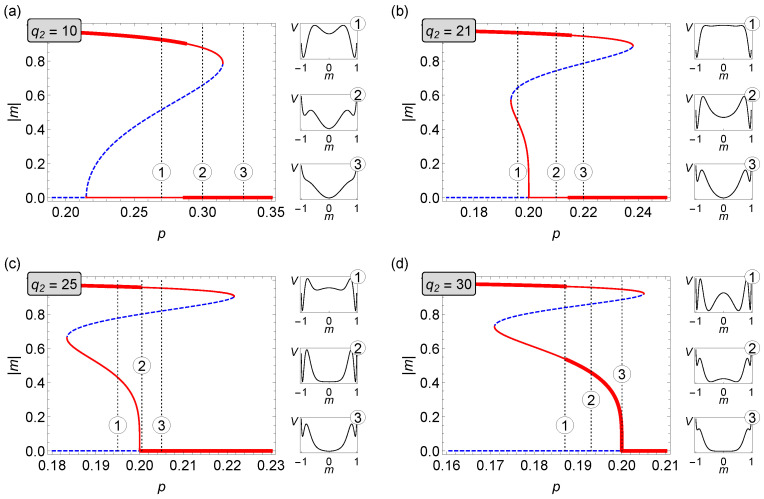
Average magnetization as a function of the parameter *p* in the asymmetric $\left(q_{1},q_{2}\right)$-voter model on duplex cliques with the LOCAL&AND rule for $q_{1}=5$ and different values of $q_{2}$. (**a**) $q_{2}=10$, (**b**) $q_{2}=21$, (**c**) $q_{2}=25$, and (**d**) $q_{2}=30$ (cf. Figure 4).

**Figure 6 entropy-22-01018-f006:**
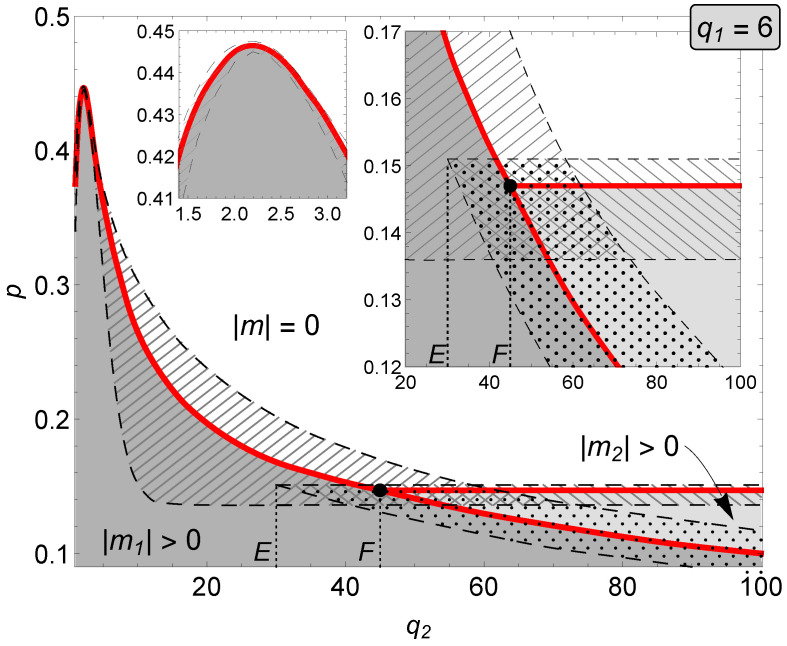
Phase diagram for the asymmetric $\left(q_{1},q_{2}\right)$-voter model on duplex cliques with the LOCAL&AND rule and $q_{1}=6$.

**Figure 7 entropy-22-01018-f007:**
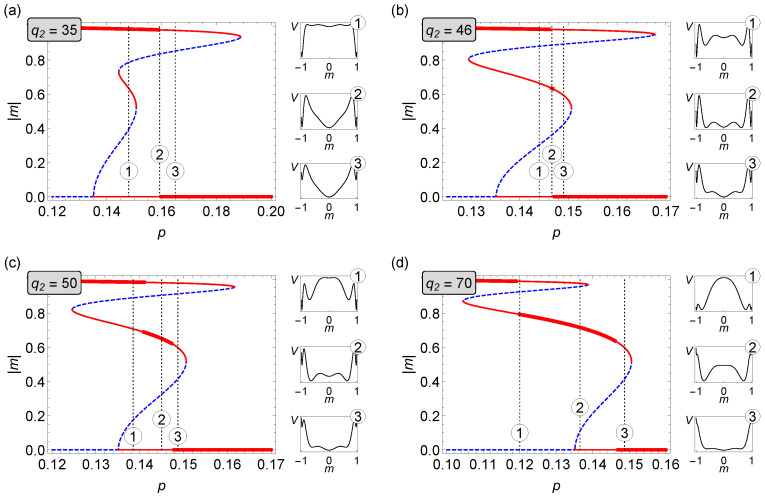
Average magnetization as a function of the parameter *p* in the asymmetric $\left(q_{1},q_{2}\right)$-voter model on duplex cliques with the LOCAL&AND rule for $q_{1}=6$ and different values of $q_{2}$. (**a**) $q_{2}=35$, (**b**) $q_{2}=46$, (**c**) $q_{2}=$0, and (**d**) $q_{2}=70$.

**Figure 8 entropy-22-01018-f008:**
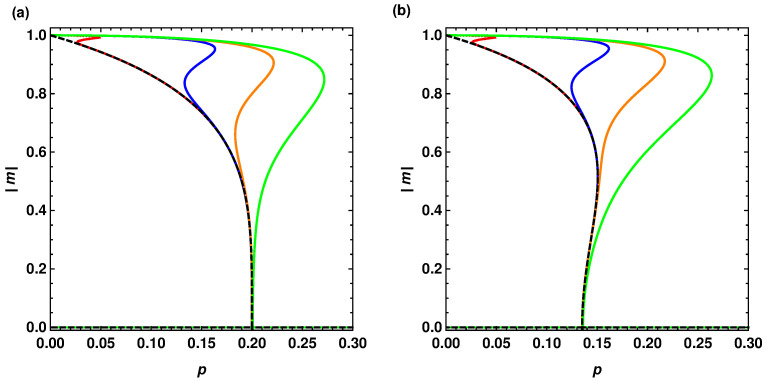
Average magnetization as a function of the parameter *p* in the asymmetric $\left(q_{1},q_{2}\right)$-voter model on duplex cliques with the LOCAL&AND rule for $q_{1}=5$ (**a**) and $q_{1}=6$ (**b**) and different values of $q_{1}$: 15 (green), 25 (orange), 50 (blue), and 500 (red). Dashed lines are solutions for the monoplex *q*-voter model obtained using Equation (Equation 8).

**Table 1 entropy-22-01018-t001:** A comparison of different versions of the *q*-voter model.

Property	Monoplex	Symmetric Duplex	Asymmetric Duplex
parameters	*p*, *q*	*p*, *q*	*p*, $q_{1}$, $q_{2}$
continuous PT	q<6	q<5	$q_{1}\leq 4$, $q_{2}\leq 4$
discontinuous PT	q≥6	q≥5	$q_{1}\leq 4$, $q_{2}\leq 4$, also observed for $q_{1}\leq 5$ and specific $q_{2}$
successive PTs	not observed	not observed	for q≤5 first discontinuous PT and then continuous one; for q≥6 two discontinuous PT
mixed-order PT	not observed	not observed	observed for $q_{1}=5$ and $q_{2}\approx 25$

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
