# Peer review of "A Veritable Zoology of Successive Phase Transitions in the Asymmetric q-Voter Model on Multiplex Networks"

_entropy, 2020, doi:10.3390/e22091018_

Round 1

Reviewer 1 Report

This manuscript provides a comprehensive study of a non-linear q voter model, discussing the features coming from different levels of complexity of this type of models, inspired in the Ising model for magnets. All the sections are carefully presented and with clear discussions and graphs. I consider that the manscript is adequate to the scope of Entropy, and I recommend its publication. I have this specific question and coment. Since the q voter model is inspired in the Ising one, it would be interesting for the reader some concise information, about the consequences of the results observed when translated to a physical system. In particular we know that phase transitions in magnetic systems are related to broken symmetries. In the figures of the manuscript there are clear depictions of these broken symmetries, that also occur in liquid crystal and adsorption models. Does these broken symmetries have a meaning in the case of the social phenomena described in the paper?. Also it would be convenient to add a table to summarize the main findings of the complex phenomena described in the text, for example, with the q system with two layers, etc. 

Author Response

We appreciate insightful comments from the reviewer. In order to summarize the main findings of this paper and compare them with previous results (obtained for monoplex and symmetric duplex) we have added a relevant table in the Conclusions section.

As it concerns the issues of broken symmetries it is hard to relate them to some specific phenomena in social sciences (with respect to successive phase transitions). Nonetheless, some classical social science experiments (S. Asch and S. Milgram) might be considered as examples of symmetry breaking and we have decided to highlight their importance in the second paragraph of the introduction section.

Reviewer 2 Report

The authors of the research paper “A veritable zoology of successive phase transitions in the asymmetric q-voter model on multiplex networks present a topic that could be relevant.

The abstract, introduction and conclusion do not fit with the conclusions.  I recommend underlining more strongly in abstract the aim and the contribution of the article to theory. Also those conclusions are not clear to me and should be rewritten.

Although many researchers have studied this topic and there are plenty of literature about it the authors should be highlight more relevant international studies to back up the paper conclusions. The lack of literature references is especially noteworthy about Monte- Carlo simulations which is a traditional topic for research.

I do suggest to the authors to rewrite the literature review including new ones and taking into account above mentioned a very deep review of international literature about the subject should be carried out.

I consider that method and empirical results have been properly realized and the research methodology is appropriate.

Finally, Please add at the end of the article the limitation of the study and the future directions of research

Author Response

We would like to thank the reviewer for his/her remarks. In order the make the aim of the study and its impact and consequences more pronounced we have respectively changed the abstract. Following we have also brought to the front the obtained results for successive phase transitions and described and discusses them in detail in conclusions. Taking into account reviewer remarks with respect to literature review we have now added a selection of international citations (9,10, 11, 16, 22, 24, 33, 34, 37) that relate to analytical and numerical (Monte-Carlo) studies in opinion formation models. Limitations and further plans of research are now placed at the very last paragraph of conclusions.

Reviewer 3 Report

The paper is well written and it brings several interesting results. However, Eq 7 is inconsistent with Eq 4. As m=2c-1, then c=(1+m)/2. In Eq 7, the Authors use c=(m-1)/2 instead. As they declare that they solve Eq 7 numerically and this solution is used for the asymmetric case, I am not able to check if their numerical results are erroneous or not. I recommend to check it and resubmit the paper.

Minor point: a misprint at the end of Eq 11, 2-nd line.

Author Response

We are most thankful for these comments: as Reviewer has pointed out Eq. (7) is in fact wrongly written by substituting c=(m-1)/2 instead of c=(m+1)/2. However, this is just a misprint that does not influence the rest of the paper, i.e., all the calculations have been performed using the proper form. Please also note that in fact Eq. (7) is connected to monoplex network and the results obtained in this study concern the asymmetric duplex case described by Eq. (11).  Moreover, if we had used the wrong substitution we couldn't have obtained proper Figures 5, 7, and 8 as in such a case magnetization is in the range <1;3> and not <-1,1>.

Following reviewer’s remark Eq. (7) has been corrected accordingly, as has also Eq. (11) (the plus sign added at the end of the second line).

Round 2

Reviewer 2 Report

The authors of the research paper “A veritable zoology of successive phase transitions in the asymmetric q-voter model on multiplex networks present a topic that could be relevant.

First of all, I am glad to see the authors have made several changes especially concerning the abstract, obtained results and discussion.

Also literature has been reviewed and added relevant one related to analytical and numerical studies.

Finally limitations and further plans, as I pointed out on my previous review, have been included.

Therefore, all my comments have been considered and so I consider this paper for publication.

Anyway for  final publication of the paper I recommend and English language minor revision.

Reviewer 3 Report

misprints:

line 31: as it was show > as it was shown
line 81: then one > than one
line 83: then it > than it
line 298: the one same > the same
line 331: phenomena > phenomena of
line 342: might > might be